# Occurrence of Potentially Toxic Elements in Bottled Drinking Water—Carcinogenic and Non-Carcinogenic Risks Assessment in Adults via Ingestion

**DOI:** 10.3390/foods11101407

**Published:** 2022-05-12

**Authors:** Elena L. Ungureanu, Alexandru D. Soare, Andreea L. Mocanu, Sorin C. Iorga, Gabriel Mustatea, Mona Elena Popa

**Affiliations:** 1National Research & Development Institute for Food Bioresources, 020323 Bucharest, Romania; elena_ungureanu93@yahoo.com (E.L.U.); alexsoare14@gmail.com (A.D.S.); andreea.mocanu1@yahoo.com (A.L.M.); soriniorga@gmail.com (S.C.I.); 2Faculty of Biotechnology, University of Agronomic Sciences and Veterinary Medicine, 011464 Bucharest, Romania; pandry2002@yahoo.com

**Keywords:** water quality, bottled water, potentially toxic elements, risk assessment

## Abstract

The presence of potentially toxic elements in drinking water can be dangerous for human health because of their bioaccumulation and toxicity, which is a huge concern for many researchers. In the case of bottled water, the exposure to toxic elements is achieved, especially by ingestion, leading to disorders of important functions of the human body. The aim of this study was the detection of some potentially toxic elements, from 50 samples of bottled drinking water, available on the Romanian market. Based on obtained concentrations, an assessment of the carcinogenic and non-carcinogenic health risk was performed. The concentrations of potentially toxic elements analyzed were below the maximum allowable limits, for all contaminants, excluding Pb and Fe. Moreover, the results of total risk via ingestion, showed that 30% of samples had an Hazard Quotient (HQ) < 1, and 70% had an HQ > 1; which implies a potential risk following the consumption of those samples. Concerning total cancer risk, 28% of the samples are in the acceptable level, while 72% of the samples are considered harmful and can lead to a type of cancer after repeated exposure. The study concluded that long term use of bottled water of poor quality may pose a hazard to human health; it is helpful for inhabitants to avoid ingestion of contaminated water.

## 1. Introduction

The quality of drinking water is a continuous concern, due to the increasing use, demand, and consumption of bottled water [1]. The most common water type used for human consumption is the tap water, available on the market in two main types of packages: glass and polyethylene terephthalate (PET) bottles. A global production of over 400 billion PET bottles is estimated each year, of which 46% of them are used for bottling drinking water, representing one of the main causes of waste production [2].

The main source of the bottled water is groundwater, representing 13–30% of the total freshwater volume, used to produce drinking water for more than 50% of the worldwide population. Groundwaters (aquifers) are complex systems, with a specific chemical composition due to the type of rocks, volcanism, evaporation processes, radioactive activities, mining activities, industrialization, and who can affect the composition of drinking water and implicitly human health and the environment. The main factor that determines the chemical contamination, of the drinking water, is the industrial pollution from natural processes and human activities such as, vehicle exhaust, poor waste disposal, fertilizer and pesticide application, untreated wastewater irrigation, bottling process, storage, and transportation [3,4]. Another important source of contamination of bottled water is through packaging material, especially PET and recycled-PET (R-PET), where a mass transfer of plastic constituents or additives from polymer to the product can occur, under the influence of certain factors, such as storage time, temperature, migrant’s solubility, and degree of plasticization. In this case, contamination with potentially toxic elements can occur due to pigments based on Pb, Cd, or Cr, catalyst residues such as antimony from antimony trioxide, a catalyst widely used to obtain PET, and the use of cadmium-based compounds as stabilizer for Polyvinyl Chloride (PVC) [4,5].

Based on their origin, bottled water can be divided into natural mineral water, spring water, and bottled drinking water. Among the physico-chemical processes to which bottled water is subjected are: de-ionization, reverse osmosis, ozone, and ultraviolet radiation [4].

An essential condition in use for human consumption, is that drinking water must be clean, free of microorganisms, parasites, or substances which, by number or concentration, may constitute a potential danger to human health [6]. Among harmful and persistent contaminants found in water, are potentially toxic metals, nitrates, and chloride [4] because water, due to its polarity, has the capacity to absorb and adsorb these contaminants [1,3]. These chemicals can accumulate in the body through direct ingestion of contaminated food, inhalation, and dermal contact, and can cause serious effects on the human body, presumably diseases of the cardiovascular, reproductive, and nervous systems, and can lead to different types of cancer [1].

Among the potentially toxic elements, As, Cd, Pb, Cr, Cu, Hg, and Ni are of major concern in research studies because these metals are presented in water in high concentration with negative health effects [7]. Some metals, such as Zn, Ni, Co, and Fe also have a structural and catalytic role for some proteins or enzymes, but in high concentrations can cause side effects [3].

The novelty and originality of the present study is based on the fact that no study has been performed on such a large number of bottled water samples, which would include both the analysis of potentially toxic elements and a complete analysis of the risk assessment. Another aspect is that the emphasis in this study was strictly on bottled water, and the chemical levels found coming from raw water contamination and migration of some metals from the packaging material.

Based on potentially toxic elements’ presence and risk of contamination, the aim of this study was the quantification of 12 toxic metals (Ba, Co, Cu, Zn, Mn, Ni, Li, Fe, Pb, Cd, Cr, and Sb) from 50 water samples available on the Romanian market. The chosen metals are regulated by the Drinking Water Quality Directives (except Ba and Li), and Annex III of the EU Regulation No. 10/2011 on plastic materials and articles intended to come into contact with food [8,9]. Based on the obtained results, a non-carcinogenic and carcinogenic risk assessment was calculated.

## 2. Materials and Methods

### 2.1. Sample Collection

In order to carry out this market study, 50 bottled drinking water samples, purchased between 2019 and 2021, were analyzed; there were five samples of carbonated mineral water and 45 samples of still mineral water. Of these samples, 36 brands of water were produced and bottled in Romania, one sample was from Fiji Islands and 13 brands came from different countries in Europe: Bulgaria (4), France (3), Italy (1), Spain (1), Island (1), Austria (1), Hungary (1), and Republic of Serbia (1). The samples tested included still natural mineral water (25), spring water (13), sparkling water (8), table water (2), semi-sparkling water (1), and artesian water (1). All the samples were stored as purchased, in PET bottles with high density polyethylene (HDPE) screw caps in laboratory conditions at room temperature until analysis.

The analyzed samples represent the main brands available on the Romanian market and were selected in order to have the same period of bottling.

### 2.2. Reagents

All the reagents used were of analytical grade. Ultrapure nitric acid (HNO_3_ 65%) was purchased from Merck (Merck Co., Darmstadt, Germany). A Multielement Standard Solution 6), used to obtain the calibration curves, was purchased from Sigma Aldrich (St. Louis, MO, USA). All dilutions were performed with ultrapure water (18.2 MΩ·cm). The glassware used for these determinations was washed, decontaminated with 10% HNO_3_ solution, and then dried in an oven at 105 °C.

### 2.3. Sample Preparation

The content of potentially toxic elements in the water samples was similar to the EPA3010A standard protocol [10], and implies evaporation to dryness of 300 mL of water, treatment of the residue with 2.5 mL of 1:1 (*v/v*) HNO_3_ solution followed by another evaporation to dryness and a second treatment with 2.5 mL of 1:1 (*v/v*) HNO_3_ solution. After heating for 15 min on a water bath, the final solution obtained was quantitatively transferred to a 50 mL volumetric flask and filled with ultrapure water.

### 2.4. Analytical Performance

The validation method was performed according to Magnusson and Ornemark, 2014 [11] and ISO/IEC 17025:2017 [12], using SPS-SW2 Surface water as certified reference material (CRM). The method shows a good accuracy with relative standard deviation of ≤4%, the recovery between 90 and 105% and a mean extended compound uncertainty (U) of 10.31%. The linearity of all calibration curves obtained (having the concentration range from 0 to 50 µg/L) was very good, with all the correlation coefficients higher than 0.996. The detection limits, obtained after 10 replicate measurements of blank solution were 0.07 μg/L for Sb and Pb; 0.09 μg/L for Ba, Cu, Mn, Ni, and Cd; 0.10 μg/L for Co and Cr; 0.11 μg/L for Li; 0.13 μg/L for Zn; and 0.15 μg/L for Fe.

### 2.5. Potentially Toxic Elements Calculation Formula

Estimation of potentially toxic elements levels was performed using Equation (1).
C = [(Cread × Df) − M) × V]/m(1)
where C is the analyte concentration in µg/L; C_read_ represents the concentration obtained from the equipment in µg/L; Df is the dilution factor (unitless); M is the concentration of blank sample in µg/L; V is the sample volume in mL; and m is the sample weight in mg.

### 2.6. Health Risk Assessment

Risk assessment is a method for evaluating the probability of occurrence of some contaminants to produce harmful health effects over a period of time. To estimate the potential carcinogenic or non-carcinogenic effects of these contaminants, a health risk evaluation is required [3].

#### 2.6.1. Non-Carcinogenic Analysis

As the main route of exposure to metals in bottled water is through ingestion, the non-carcinogenic human health risk was measured in this study through ingestion. To assess the risk caused by chronic exposure to metals, the exposure dose (D) was calculated according to Equation (2) [13]. Moreover, Hazard Quotients (HQ), hazard index (HI), and carcinogenetic risk (CR) were calculated according to the model used by Mohammadi et al., 2019 [3]. These parameters were established for an adult with an average weight of 70 kg and a water intake rate of 2 L/day.
D = (C × IR × EF)/BW,(2)
where D is the exposure dose (mg/kg/day), C is the contaminant concentration (mg/L), IR is the intake rate of contaminated water (L/day), EF is the exposure factor (unitless), BW is the body weight (kg).

The HQ for each metal was calculated by reporting the exposure dose (D) to the oral reference dose (RfD) for the same heavy metal. This parameter was calculated using Equation (3).

The values of RfD for oral ingestion for Pb, Cd, Cr, Co, Ni, Zn, and Cu are 3.6 × 10^−3^ mg/kg/day, 5 × 10^−4^ mg/kg/day, 3 × 10^−3^ mg/kg/day, 2 × 10^−2^ mg/kg/day, 2 × 10^−2^ mg/kg/day, 3 × 10^−1^ mg/kg/day, and 3.7 × 10^−2^ mg/kg/day, respectively [14]. For Ba, Fe, Mn, and Sb, the values of reference dose are 7 × 10^−2^ mg/kg/day [3], 7 × 10^−3^ mg/kg/day [15], 4.7 × 10^−2^ mg/kg/day [16], 2.8 × 10^−2^ mg/kg/day [17] and 3.5 × 10^−4^ mg/kg/day, respectively [18].
HQ = D/RfD(3)
where HQ is the hazard quotient (unitless); D is the exposure dose (mg/kg/day); and RfD is the reference dose (mg/kg/day), which represents the tolerable daily intake of the metal via oral exposure.

By summing the HQ values for all tested metals, hazard index (HI) is obtained, according to Equation (4). The HI represents the total potential non-carcinogenic health risk caused by the metals present in tested waters [19].
HI = HQ_Ba_ + HQ_Co_ + HQ_Cu_ + HQ_Zn_ + HQ_Mn_ + HQ_Ni_ + HQ_Li_ + HQ_Fe_ + HQ_Pb_ + HQ_Cd_ + HQ_Cr_ + HQ_Sb_
(4)
where HI is the hazard index (unitless), and HQ is the hazard quotient (unitless) of all potentially toxic elements tested.

#### 2.6.2. Carcinogenic Analysis

Carcinogenic risk (CR) represents the probability of a person developing any form of cancer throughout life, by exposure to a contaminant 24 h a day, for 70 years (in the case of an adult) [20]. This parameter was evaluated for Pb, Cd, Cr, and Ni, via ingestion pathway, using Equation (5).
CR = D × CSF(5)
where CR is the cancer risk (unitless); D is the exposure dose in mg/kg/day; and CSF is the Cancer Slope Factor, in mg/kg/day. The values of CSF for oral ingestion are 8.4 × 10^−1^ (Ni), 8.5 (Pb), 6.1 (Cd), and 41 (Cr) [3].

### 2.7. Data Analysis

All experiments were conducted in triplicate, and results were expressed as mean of the replicated ± standard deviation. SPSS Statistics, version 20.0 (SPSS Inc., Chicago, IL, USA) was used for the statistical analyses, to identify the relationships between tested elements and between potentially toxic elements and source of extraction using a one-way analysis of variance (ANOVA).

## 3. Results and Discussions

### 3.1. Potentially Toxic Elements Concentrations in Bottled Water

The concentrations of analyzed potentially toxic elements are presented in Table 1 as maximum and minimum values and the mean of the triplicate analysis.

The ascendent trend of toxic metals concentrations, based on mean values, is Cd > Sb > Co > Cr > Mn > Pb > Cu > Ni > Li > Ba > Zn > Fe.

Most likely, the differences between the results are due to the different characteristics of the aquifer (drilling depth, altitude of the capture spring, and characteristics of the rock layers through which the spring crosses). Zakir et al., 2020 [19] reached the same assumption in their study.

Regarding the detection rate of the tested metals in the samples, these decrease as following, Cu, Zn, Ni, and Fe: (100% of samples) > Pb (96%) > Li (64%) > Mn (58%) > Co (56%) > Sb (50%) > Ba (40%) > Cr (38%) > Cd (0%).

The values found were compared with the values imposed by the Directive (EU) 2020/2184 [9]. Lead and iron registered exceeding of the imposed limits, of 5 µg/L and 200 µg/L, respectively; in the case of Pb, only one sample exceeded the limit, while in the case of Fe there were 14 samples exceeding the limit.

As can be seen from Table 2, the values obtained in the present study were compared with those available in the literature, which obtained higher or lower concentrations.

Regarding Ba, in the study by Wu et al., 2018 [21] a value of 384 µg/L was found, quite high, even if this metal does not have an imposed limit. Concerning Co, lower values were obtained in the study conducted by Wu et al., 2018 [19] and higher values were obtained by Bakirdere et al., 2013 [22].

In the case of Cu, the levels found in other studied articles [23,24] were relatively low or not detected [25,26,27]. Values higher than those obtained in the present study can be observed in the studies conducted by Hadiani et al., 2015; Ristic et al., 2011; Wu et al., 2018; and Salmani et al., 2017 [21,28,29,30], but not exceeding the limit of 100 ppb.

Concerning Zn, in research by Singla et al., 2014; Gautam, 2020; and Bamuwamye et al., 2017 [25,27,31], this trace heavy metal was not detected, but higher concentrations were obtained by Salmani et al., 2017 [30] and Wu et al., 2018 (2260 µg/L) [21] without exceeding the required limit.

Regarding Mn, in the study by Bamuwamye et al., 2017 [27] it was not detected, but Ristic et al., 2011 [29] obtained values of 195.2 µg/L, which are much higher than the limit of 50 µg/L imposed by the European Directive [8]. Lower levels of Ni were obtained by Ristic et al., 2011 [29] and Bamuwamye et al., 2017 [27], and higher by Gautam, 2020; Wu et al., 2018; Salmani et al., 2017; and Fakri et al., 2015 [21,30,31,32], but lower than maximum imposed limit of 20 µg/L.

Concerning Fe, the levels obtained by Salmani et al., 2017 and Gautam, 2020 [30,31] were lower than those obtained in the present study, even undetectable in the case of Bamuwamye et al., 2017 [27], but similar to those obtained by Maxwell et al., 2018 and Singla et al., 2014 [25,33].

If we report to the European Directive [8], in the case of Pb, the highest obtained concentrations were 348 µg/L [33], 320 µg/L [25], 241 µg/L [27], 210 µg/L [24], 15.5 µg/L [21], and 12.66 µg/L [22], values which exceed the maximum allowable limit. There have also been studies [24,29,30,31], where the concentrations obtained were lower than those in this study, and even undetectable in the study by Gautam, 2020 [31].

Concerning Cd, similar levels were obtained by Anibaldi et al., 2018; Salmani et al., 2017; Gautam, 2020; Bakirdere et al., 2013; and Bamuwamye et al., 2017 [22,23,27,30,31], but also higher concentration than those of present study were found in the studies of Hadiani et al., 2015; Sarvestani and Aghasi, 2019; Ristic et al., 2011; Wu et al., 2018; and Fakri et al., 2015 [21,24,28,29,32], but without exceeding the imposed limit. Instead, the samples analyzed by Maxwell et al., 2018 [32] recorded values of 72.1 µg/L, which far exceed the limit imposed by the European Directive [8].

**Table 2 foods-11-01407-t002:** Comparative study related to potentially toxic elements concentration in bottled drinking water.

Element Analyzed	Number of Samples	Concentration Range (µg/L)	Analytical Method	Reference
Ba, Co, Cu, Zn, Mn, Ni, Li, Fe, Pb, Cd, Cr, Sb	50	<0.09–10.47 ± 6.79; <0.10–0.89 ± 0.75; 0.35 ± 3.78–5.63 ± 9.57; 0.67 ± 9.86–15.20 ± 5.29; <0.09–7.41 ± 1.55; 0.16 ± 1.04–3.77 ± 1.91; <0.11–12.30 ± 9.38; 13.73 ± 7.49–1761.24 ± 1.23; <0.07–6.0 ± 0.25; <0.09; <0.10–4.02 ± 2.12; <0.07–0.64 ± 6.85	ICP–MS	Present study
Pb, Cd, Cu, As, Hg	42	<3–5.1 ± 0.6; <0.6–1.2 ± 0.2; <3–19.7 ± 2.6; <3–7.9 ± 0.4; <0.3–0.6 ± 0.1	GF–AAS MHS–AAS	[28]
Cd, Pb, Cu	23	0.0008–0.0024; 0.006–0.025; 0.083–0.37	(SWASV)	[23]
Pb, Cu, Cd	-	0.1–210; LOD–3; LOD–2	GF–AAS	[24]
As, Cd, Cr, Cu, Mn, Ni, Pb, Sb	23	<0.20–6.41; <0.01–1.19; <0.21–1.57; 0.11–9.50; 0.04–195.2; <0.30–8.38; <0.04–2.89; <0.03–1.81	ICP–MS	[29]
Ag, As, Ba, Be, Cd, Co, Cr, Cu, Mo, Ni, Pb, Sb, Sn, Se, Tl, U, V, Zn	59	<0.0004–0.018; 0.025–99.6; 15.3–384; <0.002–0.700; <0.001–2.17; <0.002–3.08; <0.02–4.14; <0.02–18.1; <0.003–10.0; <0.01–15.2; <0.004–15.5; <0.001–0.653; <0.001–1.41; 5.57–1220; <0.001–0.141; 0.002–5.41; <0.01–78.7; <0.05–2260	ICP–QMS	[21]
Cu, Zn, Ni, Fe, Al, Pb, Cd	2	34.5–35.4; 19.0–32.1; 1.3–4.1; 4.5–5; 12.6–15.3; 3.1–3.2; <0.0002	AAS	[30]
Cd	8	LOD–2.1 ± 0.22	AAS	[32]
Cd, Cr, Pb, As, Ni, Fe	20	LOD–72.1; LOD; 0–348.1; LOD; LOD–12.4; LOD–71.5	GF–AAS	[33]
Fe, Cu, Pb, Se, Zn, Cr, B, Al	4	28–40; 0; 8–320; 30–3290; 0; 0; 280–540; 0	ICP–AES	[25]
Cu	20	0.03–1.71	ICP–MS	[26]
Fe, Zn, Ag, Cu, Cd, Co, Ni, Pb	200	3–3.7; ND; ND; 8.4–10.5; ND; 27.2–28.3; 7.6–8.9; ND	F–AAS	[31]
As, Cd, Pb	78	<2–11.54 ± 2.79; <0.036; <0.25–12.66 ± 0.68	GF–AAS	[22]
Al, As, Cd, Cr, Cu, Fe, Hg, Mn, Ni, Pb, Zn	9	ND; ND; ND; ND–107; ND; ND; ND; ND; ND; 91–241; ND	GF–AAS	[27]

ICP–MS: inductively coupled plasma mass spectrometry; GF–AAS: graphite furnace atomic absorption spectrometry; MHS–AAS: atomic absorption spectrometer equipped with hydride system; SWASV: square wave anodic stripping voltammetry; ICP–QMS: Inductively coupled plasma–quadrupole mass spectrometer; AAS: atomic absorption spectrometry; ICP–AES: inductively coupled plasma–atomic emission spectroscopy; F–AAS: flame–atomic absorption spectrometry; LOD: limit of detection; ND: not detectable.

Cr levels were lower in the study conducted by Ristic et al., 2011 [29] and even undetectable in samples analyzed by Maxwell et al., 2018 and Singla et al., 2014 [25,33]. Higher concentrations than those obtained in the present study were in the samples tested by Wu et al., 2018 [21]. Concentrations higher than the European limit of 25 µg/L were obtained by Bamuwamye et al., 2017 [27], which was of 107 µg/L.

Ristic et al., 2011 [29] obtained values of Sb higher than those from this study, but lower than the limit of 10 µg/L imposed by European Directive [8].

The values found for Pb, Cd, and Cr in the studied articles were much higher compared with the limits imposed by the European Directive, especially in the case of Pb. This may be due to the geological contamination of the raw water, contamination of water during distribution through iron piper [34], or also the migration of some metals from the packaging material under certain storage conditions, including temperature and storage time [35]. Small differences can also occur due to the analytical method used to determine the content of metals.

Potentially toxic elements become toxic when they accumulate in the tissue and are not metabolized by the body. The main pathway of exposure to heavy metals for nonoccupational workers occurs via ingestion of food and contaminated water, but also through inhalation or dermal contact with products containing toxic metals [36].

Ba poisoning is manifested by blocking of potassium channels, without affecting the NA/K pump, which can lead to increasing of intracellular potassium and extracellular hypokalemia. Moreover, after exposure to high levels of Ba, vomiting and diarrhea can appear, but also, dilated pupils, slow pulse, tingling of the mouth, neck, and limbs, and respiratory problems [37].

Co is a trace mineral which is an integral part of vitamin B12, involved in proper thyroid function and blood pressure regulation [38]. Other important functions of Co in the body are the production and regulation of red cells and Deoxyribonucleic acid (DNA), synthesis of fatty acids, and energy production. However, in the case of humans exposed to excessive levels of Co, some systemic toxic effects can appear, including peripheral neuropathy, hearing and vision loss, cognitive decline, cardiomyopathy, hypothyroidism, polycythemia, weakness, fatigue, and neuromuscular symptoms such as decreased muscle mass, tremor, and convulsions, which are quite rare [39].

Cu is a trace mineral involved in biological systems because it is an integral component of many enzymes, such as ceruloplasmin, cytochrome c oxidase, superoxide dismutase, and tyrosinase [40]. In humans, copper toxicity includes gastrointestinal symptoms, such as nausea, vomiting, abdominal pain, stomach irritation, and liver enzyme disfunction [41]. Studies have demonstrated that Cu is involved in neurodegenerative disease such as Alzheimer, Parkinson, and Huntington disease, as well as other amyotrophic lateral sclerosis [42].

Zn, which is an important cofactor in the body, being essential for its normal function, can become toxic in the case of increased levels. Acute toxic ingestion can cause gastrointestinal symptoms (hematemesis), renal injury (asymptomatic hematuria, interstitial nephritis, and acute tubular necrosis), but also liver necrosis, thrombocytopenia, coagulopathy, and even death. Chronic exposure to Zn can produce bone marrow, neurological effects, gastrointestinal problems, sideroblastic anemia, granulocytopenia, myelodysplastic syndrome, and ascending sensorimotor polyneuropathy syndrome [43]. An excessive intake of this element may play the main role in cancer etiology [44].

Mn is considered a trace element for the normal functioning of the body because it is implicated in many important functions such as bones and brain development. However, its presence in excess can lead to disorders which depend upon the route of exposure. Air inhalation can produce neurological disorders, such as Parkinson disease, loss of memory, anxiety, and sleep problems; while through ingestion, it can cause accumulation of Mn in blood and brain, which can lead to neurological problems. Drinking water contaminated with Mn produces neurological disorders, especially in babies and children [45]. Both children and adults who cannot remove the excess of Mn from the body can develop nervous system problems [44].

Regarding Ni, which is not an essential element but is involved in red blood cell synthesis and activation of some enzyme system [38], chronic exposure to Ni can produce dermatitis and allergy, skin and oral epithelium damage, lung cancer and nasal cancer (for refinery workers), respiratory problems, decreased body weight, and nervous system damage [38,46]. Exposure to this heavy metal has been associated with lymphoblastic leukemia and exocrine pancreatic cancer [47]. The target organs in the case of exposure to high levels of Ni are kidneys, brain, liver, bones, lungs, and endocrine system [45].

Exposure to high levels of Li in the long term can cause adverse effects on tree organ systems, such as thyroid gland, kidney, and parathyroid gland, but the most important factor due to the prevalence of Li is hypothyroidism. The absorption of this metal is through gastrointestinal tract and circulated by the blood through the body [40].

Although Fe is a trace element involved in many vital functions and structures, after exposure to high levels, it can produce gastrointestinal bleeding, vomiting, diarrhea, gastrointestinal ulcerations, tachycardia, hypertension, hepatic necrosis, metabolic acidosis, and lung cancer [48,49].

In the case of Pb, the research studies have shown a strong correlation between this metal and kidney cancer, exocrine pancreatic cancer, increased risk of developing renal cell carcinoma, and carcinogenesis in lung tissue [47]. Other side effects caused by exposure to high lead concentrations are high risk of hypertension, gastrointestinal diseases, neurological disorders, thyrotoxicity, and disorders of reproductive system such as reduction in libido, sperms count and their mobility, leading to infertility. Moreover, this metal can pass placenta and affect a fetus, causing lower IQ level, encephalopathy, and neurological disorders [45].

Among the side effects that can occur with a chronic intake of Cd are lung cancer, kidney damage, fragile bones [19,48], bronchitis, damage to DNA and the immune system and inhibiting DNA repair processes and apoptosis [43], disruptions to the ovarian function by mimicking endogenous estrogen which can produce an increased risk of ovarian and breast cancer [46]. Long-term exposure can cause hypertension, arthritis, anemia, diabetes, osteoporosis, cardiovascular diseases, and cancer [48]; it is classified in Group I carcinogens for humans by International Agency for Research and Cancer (IARC) [49].

Cr is an important trace element involved in controlling blood sugar and lipid levels [45], but in high concentration of its trivalent oxidation state, can lead to lung cancer and cellular damage, while the hexavalent form can cause blood cell damage by oxidation, functional degradation of the liver and kidney [46], ulcers, corrosive reactions on the nasal spectrum, acute irritation dermatitis, and allergic dermatitis [50].

Chronic exposure to Sb compounds can cause respiratory effects such as pneumoconiosis, chronic bronchitis, inactive tuberculosis, emphysema, respiratory irritation, and cardiovascular disorders such as increased blood pressure, altered electrocardiography, degenerative changes in the myocardium, and gastrointestinal effects including abdominal pain, diarrhea, vomiting, and ulcers. Sb poisoning can also produce skin problems, such as pustules, and eruptions in the trunk and limbs and reproductive disfunctions characterized by perturbances in menstruation and even spontaneous abortion. The genotoxicity and carcinogenicity of Sb manifests through chromosome breakage, increased oxidative damage to DNA, chromosomal aberration, but also the possibility to develop lung tumors in rats, especially in the case of antimony trioxide which is classified as possibly carcinogenic to humans (Group 2B) by IARC [51].

### 3.2. Health Risk Assessment

#### 3.2.1. Non-Carcinogenic Analysis

Table 3 summarizes the calculated exposure dose (D) values for consumption of drinking water. The ascendent trend for exposure dose, based on the content of potentially toxic elements, is the following: Cd > Sb > Co > Cr > Mn > Pb > Ni > Cu > Li > Ba > Zn > Fe. HQ values, which are presented in the same table, decreased in the order: Cd > Co > Mn > Zn > Ba > Li > Ni > Cr > Pb > Sb > Cu > Fe.

If the HQ result is less than or equal to 1, repeated exposure may not cause side effects, but if the value is greater than 1, then consumers are exposed to a potential risk [52]. In our case, it can be seen that only in the case of Fe, there was evidence that HQ was greater than 1. Thus, out of the total of 50 samples, 32% had an HQ < 1, the rest of the samples had an HQ > 1, which implies a potential risk on consumers in the case of chronic consumption.

Moreover, to estimate the total potential non-carcinogenic risk in the potentially toxic elements tested the hazard index (HI) was calculated. The values were between 1.38 × 10^−2^ and 7.17, with an average value of 1.89. As in the case of HQ, if HI is greater than 1, there is a possibility that the metals may produce some adverse effects, but non-carcinogenic, after chronic ingestion of the contaminant. If HI is less than or equal to 1, then no side effects will occur after chronic exposure [3]. Of the 50 samples, 30% have and HI greater than 1, which implies the occurrence of adverse effects following the consumption of those samples (Figure 1).

The hazard index values via ingestion obtained in other studies varied between 3.65 × 10^−1^ and 2.04 (Duggal et al., 2018) [53], 2.44 × 10^−1^ and 2 × 10^−2^ [54], 3.9 × 10^−4^ and 1.31 × 10^−2^ [19], and 2.13 × 10^−4^ and 1.54 × 10^−1^ [55]. As observed, in the study by Duggal and Rani, 2018 [53], HI values were higher than 1, which implies that by repeated exposure consumers may develop side effects.

#### 3.2.2. Carcinogenic Analysis

Because Pb, Cd, Cr (VI), and Ni can potentially enhance the risk of cancer in humans after long term exposure to low amounts of toxic metals, the carcinogenic risk analysis was calculated only for these metals.

If for one heavy metal, CR is less than 1 × 10^−6^, it is considered tolerable; a value between 1 × 10^−6^ and 1 × 10^−4^ is considered an acceptable range; and higher than 1 × 10^−4^ is considered intolerable or harmful to human health [52].

In the case of Pb, 4% of samples were tolerable, 94% were intolerable range and only 2% were considered intolerable. For Cd all the samples were tolerable because this metal was not detected in any sample. Concerning Cr (VI), 58% of the samples were considered tolerable, 34% were in an acceptable range; and 8% were intolerable. Regarding Ni, all the samples were within the acceptable range.

The ascendent trend of CR, based on Cancer Slope Factor and potentially toxic elements concentrations, is: Cd > Ni > Pb > Cr.

For the fourth metals, the total CR was calculated, which ranged between 4 × 10^−4^ and 4.86 × 10^−3^, with a mean value of 5.14 × 10^−4^. A value less than 1 × 10^−6^ is considered insignificant, while a value above 1 × 10^−4^ is considered harmful to human health. The acceptable level for total CR, for each exposure pathway (ingestion and dermal), is 1 × 10^−5^ [3].

Out of a total of 50 samples, 28% of the samples were in the acceptable level, while 72% of the samples were considered harmful, which can lead to a type of cancer (Figure 2).

The total cancer risk values via ingestion obtained in other research studies varied between 1.77 × 10^−6^ and 2.03 × 10^−5^ [19], 1.65 × 10^−6^ and 8.05 × 10^−5^ [54], and 1.47 × 10^−2^ and 5.05 × 10^−4^ [3]. As observed, the samples tested by Mohammadi et al., 2019 [3] were considered harmful to human health after repeated exposure.

### 3.3. Statistical Analysis

The correlation study was carried out to establish the relationship among tested potentially toxic elements, but also between these metals and the region of extraction. The aim of this statistical analysis was to discover if the presence of a particular metal facilitates the occurrence of others potentially toxic elements, or if the presence of contaminants is due to the anthropogenic activity of the region of extraction. Concerning correlation between metals, the parameters were moderate and correlated with each other at *p* < 0.01 and *p* < 0.05.

Positive correlation was present in some heavy metal pairs, such as Ba–Fe (r = 0.349), Co–Cu (r = 0.578), Co–Ni (r = 0.740), Co–Fe (r = 0.538), Cu–Zn (r = 0.370), Cu–Ni (r = 0.457), Zn–Cu (r = 0.370), Ni–Fe (r = 0.681), Fe–Sb (r = 0.386), Cr–Sb (0.804). Regarding Mn and Li, they were not correlated with any metal. The absence of correlations between toxic metals and the region proves that the tested samples do not have close extraction sources. Most likely, the lack of correlations between the analyzed metals demonstrates that their content in the tested samples is not due to a single factor but to several factors related to the soil structure and the anthropogenic activity in the area. The statistical analysis performed on the samples tested by Hussain et al., 2019 [56], led to a similar conclusion.

The results obtained demonstrated that there was no correlation between metals and the water extraction area. The positive correlation between the potentially toxic elements tested may suggest the possibility that, for some samples, the extraction sources may be quite close to each other.

Regarding sigma two-tailed values, in the case of Ba, Mn, Li, and Pb, all values were >0.05, which means that there is no statistically significant correlation between these metals and the rest of the metals tested. In the case of Co, there was no significant correlation with Ba, Zn, Mn, Li, Pb, Cr, and Sb (values > 0.05), but a significant correlation with Cu, Ni, and Fe, where values were <0.05. Cu was corelated with Co, Zn, and Ni; Zn was correlated only with Cu (value 0.026); Ni with Co, Cu, and Fe; Fe with Ba, Co, Ni, and Sb; Cr only with Sb; and Sb with Fe and Cr.

## 4. Conclusions

The present study was conducted in order to evaluate the concentrations of 12 potentially toxic elements, but also the carcinogenic and non-carcinogenic health risk assessment of these metals in 50 samples of bottled drinking water, available on the Romanian market. The results show that the concentration of all tested elements, except Fe and Pb, were below the maximum allowable limits, and from this point of view, represent a cause for concern among consumers. The order of heavy metal toxicity based on mean concentrations is: Cd > Sb > Co > Cr > Mn > Pb > Cu > Ni > Li > Ba > Zn > Fe. The ascendent trend for exposure dose, based on the content of potentially toxic elements, is the following: Cd > Sb > Co > Cr > Mn > Pb > Ni > Cu > Li > Ba > Zn > Fe. Out of the total of 50 samples, 32% had an HQ < 1, the rest of the samples had an HQ > 1, which implies a potential risk for consumers in the case of chronic consumption. The calculated HI_oral_ showed that 30% of samples have an HI greater than 1, which implies the occurrence of adverse effects following the consumption of those samples. Regarding carcinogenic health risk, 28% of the tested samples were in the acceptable range, while 72% of the samples were considered harmful, which can lead to a type of cancer. Finally, it was concluded that the drinking water does not represent a chronic health risk, but if potentially toxic elements concentrations exceed the safe level, side effects may occur due to consumption of contaminated water. To avoid ingesting large amounts of metals, bottled water can be filtered before consumption. Much more detailed studies are also needed to determine the exact degree of contamination of bottled water and ways to reduce the risk of ingesting large amounts of potentially toxic elements.

## Figures and Tables

**Figure 1 foods-11-01407-f001:**
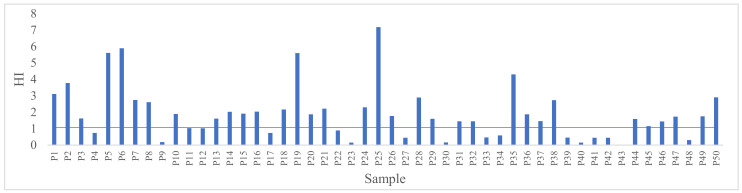
HI values of the tested samples.

**Figure 2 foods-11-01407-f002:**
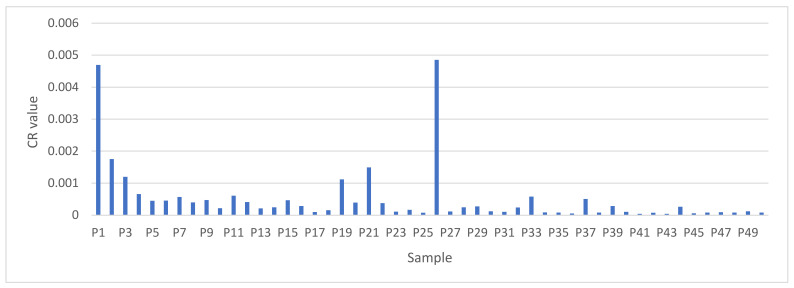
Total cancer risk values of the tested samples.

**Table 1 foods-11-01407-t001:** Potentially toxic elements levels in drinking water samples.

Element	Potentially Toxic Elements Concentrations (µg/L)			Directive EU 2020/184 (µg/L)
	Min	Max	Mean	
Ba	<0.09	10.47 ± 6.79	1.90 ± 161.41	-
Co	<0.10	0.89 ± 0.75	0.11 ± 147.59	-
Cu	0.35 ± 3.78	5.63 ± 9.57	1.09 ± 90.02	2000
Zn	0.67 ± 9.86	15.20 ± 5.29	3.17 ± 75.75	-
Mn	<0.09	7.41 ± 1.55	0.42 ± 269.11	50
Ni	0.16 ± 1.04	3.77 ± 1.91	1.21 ± 73.47	20
Li	<0.11	12.30 ± 9.38	1.67 ± 149.53	-
Fe	13.73 ± 7.49	1761.24 ± 1.23	455.76 ± 85.70	200
Pb	<0.07	6.0 ± 0.25	0.59 ± 150.33	5
Cd	<0.09	<0.09	-	5
Cr	<0.10	4.02 ± 2.12	0.28 ± 287.21	25
Sb	<0.07	0.64 ± 6.85	0.09 ± 146.40	10

**Table 3 foods-11-01407-t003:** Risk assessment of potentially toxic elements levels in drinking water samples.

Element	Exposure Dose (µg·kg^−1^·day^−1^)			Hazard Quotient (HQ)			Cancer Risk (CR)		
	Max	Min	Mean	Max	Min	Mean	Max	Min	Mean
Ba	2.99 × 10^−4^	NA	5.45 × 10^−5^	4.27 × 10^−3^	NA	7.79 × 10^−4^	-	-	-
Co	2.54 × 10^−5^	NA	3.19 × 10^−6^	1.27 × 10^−3^	NA	1.59 × 10^−4^	-	-	-
Cu	5.54 × 10^−4^	1.0 × 10^−5^	4.13 × 10^−5^	1.50 × 10^−1^	2.70 × 10^−3^	1.11 × 10^−2^	-	-	-
Zn	4.34 × 10^−4^	1.91 × 10^−5^	9.04 × 10^−5^	1.45 × 10^−3^	6.38 × 10^−5^	3.01 × 10^−4^	-	-	-
Mn	2.12 × 10^−4^	NA	1.17 × 10^−5^	4.60 × 10^−3^	NA	2.55 × 10^−4^	-	-	-
Ni	1.08 × 10^−4^	3.71 × 10^−6^	3.46 × 10^−5^	5.39 × 10^−3^	1.86 10^−4^	1.73 × 10^−3^	9.05 × 10^−5^	3.21 × 10^−6^	2.90 × 10^−5^
Li	3.51 × 10^−4^	NA	4.77 × 10^−5^	1.26 × 10^−2^	NA	1.70 × 10^−3^	-	-	-
Fe	5.0 × 10^−2^	NA	1.30 × 10^−2^	7.17	NA	1.86	-	-	-
Pb	1.71 × 10^−4^	NA	1.85 × 10^−5^	4.76 × 10^−2^	NA	5.13 × 10^−3^	1.46 × 10^−3^	NA	1.57 × 10^−4^
Cd	NA	NA	NA	NA	NA	NA	NA	NA	NA
Cr	1.15 × 10^−4^	NA	8.01 × 10^−6^	3.83 × 10^−2^	NA	2.67 × 10^−3^	4.71 × 10^−3^	NA	3.28 × 10^−4^

NA: not applicable.

## Data Availability

Not applicable.

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
