# Peer review of "Occurrence of Potentially Toxic Elements in Bottled Drinking Water—Carcinogenic and Non-Carcinogenic Risks Assessment in Adults via Ingestion"

_foods, 2022, doi:10.3390/foods11101407_

Round 1

Reviewer 1 Report

File attached. 

Reviewer 2 Report

Comments:

  1. Use Keywords as “Keywords: Water quality; bottled water; heavy metals; risk assessment”
  2. Add a unit for values given in the sentence: “The values of RfD for oral ingestion for Pb, Cd, Cr, Co, Ni, Zn and Cu are 3.60E-03 5.00E-04, 3.00E-03, 2.00E-02, 2.00E-02, 3.00E-01, respectively 3.70E-02 [11]. For Ba, Fe, Mn and Sb, the values of reference dose are 7.00E-02 [3], 7.00E-03 [12], 4.70E-02 [13], 2.80E-02[14] and 3.50E-04 [15].
  3. Equation 5 is not true. Use “×” instead “/”.
  4. To improve the Introduction or Discussion, the authors can read and use the following papers:

-Non-carcinogenic health risk assessment of nitrate in bottled drinking waters sold in Iranian markets: a Monte Carlo simulation

Reviewer 3 Report

This manuscript reports the occurrence of heavy metals in bottled drinking water – carcinogenic and non-carcinogenic risk assessment in adults via ingestion.

- English writing needs further polish.

- The title of the manuscript must be changed to "Occurrence of heavy metals in bottled drinking water – carcinogenic and non-carcinogenic risks assessment in adults via ingestion".

- The "conclusion" section in the abstract part must be improved.

- The keywords provided by the authors are mainly derived from the main title. The authors should try to provide some different keywords. This would increase the visibility of the paper by search engines if accepted for publication by the journal.

- The section "Introduction" is not well organized to reveal the importance of the work. It does not the sufficient solidarity. Therefore, the structure and logic of the introduction need to be modified.

- Adverse health effects of studied elements must be presented in the introduction section. In so doing, it is suggested that the following articles be used as a reference:

i) Marine Pollution Bulletin, 123(1-2): 34-38 (2017).

ii) Environmental Science and Pollution Research, 25(3): 2664-2671 (2018).

iii) Biological Trace Element Research, 187(2): 602-610 (2019).

- The authors should try to bring out what is the originality of the work done and why the results may have a particular interest for the scientific community. 

- The novelty is not sufficiently explained or clear in the introduction section. Also, the research gap should be clearly described.

- The objectives were too general. Therefore, this part of the manuscript must be rewritten to be more detailed.

- It seems that the RfD values of some elements (e.x. Cd and Cr) are incorrect.

- To improve the manuscript quality, the carcinogenic and non-carcinogenic risks of children must be computed and discussed.

- The quality of the discussion section must be improved. In so doing, organize the discussion from the general to the specific, linking your findings to the literature, then to theory, then practice and avoid repetition from the introduction.

Round 2

Reviewer 1 Report

Accept in present form

Reviewer 3 Report

The authors have revised their manuscript and responded to all the points that I had raised in the previous version of the manuscript. Therefore, the manuscript can be accepted for publication in Foods as it is.